# Interplay between Coumarin Accumulation, Iron Deficiency and Plant Resistance to *Dickeya* spp.

**DOI:** 10.3390/ijms22126449

**Published:** 2021-06-16

**Authors:** Izabela Perkowska, Marta Potrykus, Joanna Siwinska, Dominika Siudem, Ewa Lojkowska, Anna Ihnatowicz

**Affiliations:** 1Laboratory of Plant Protection and Biotechnology, Intercollegiate Faculty of Biotechnology of University of Gdansk and Medical University of Gdansk, University of Gdansk, Abrahama 58, 80-307 Gdansk, Poland; izabela.perkowska@phdstud.ug.edu.pl (I.P.); marta.potrykus@gumed.edu.pl (M.P.); siwinskajoanna@gmail.com (J.S.); d.siudem@gmail.com (D.S.); ewa.lojkowska@biotech.ug.edu.pl (E.L.); 2Department of Environmental Toxicology, Faculty of Health Sciences with Institute of Maritime and Tropical Medicine, Medical University of Gdansk, Debowa 23 A, 80-204 Gdansk, Poland

**Keywords:** abiotic stress, biotic stress, fraxetin, iron deficiency, scopoletin, pathogen, plant–environment interactions, mineral nutrition

## Abstract

Coumarins belong to a group of secondary metabolites well known for their high biological activities including antibacterial and antifungal properties. Recently, an important role of coumarins in plant resistance to pathogens and their release into the rhizosphere upon pathogen infection was discovered. It is also well documented that coumarins play a crucial role in the *Arabidopsis thaliana* growth under Fe-limited conditions. However, the mechanisms underlying interplay between plant resistance, accumulation of coumarins and Fe status, remain largely unknown. In this work, we investigated the effect of both mentioned factors on the disease severity using the model system of Arabidopsis/*Dickeya* spp. molecular interactions. We evaluated the disease symptoms in Arabidopsis plants, wild-type Col-0 and its mutants defective in coumarin accumulation, grown in hydroponic cultures with contrasting Fe regimes and in soil mixes. Under all tested conditions, Arabidopsis plants inoculated with *Dickeya solani* IFB0099 strain developed more severe disease symptoms compared to lines inoculated with *Dickeya dadantii* 3937. We also showed that the expression of genes encoding plant stress markers were strongly affected by *D. solani* IFB0099 infection. Interestingly, the response of plants to *D. dadantii* 3937 infection was genotype-dependent in Fe-deficient hydroponic solution.

## 1. Introduction

The secretion of phenolic compounds from roots into the rhizosphere has long been recognised as a component of the acidification-reduction strategy to acquire iron (Fe), occurring in all plant species except grasses [1]. However, the molecular mechanisms underlying these processes remained elusive until recently, when several research groups including our team, independently demonstrated the important role of plant secondary metabolites called coumarins for the growth of a model plant *Arabidopsis thaliana* (hereafter Arabidopsis) under Fe-limited conditions [2,3,4,5,6,7,8,9]. It was proven that coumarins are involved in Fe chelation and that secretion of coumarins by Arabidopsis roots is induced under Fe-deficiency. The biological roles of novel enzymes involved in coumarin biosynthesis, which in parallel maintain Fe homeostasis in plants, were elucidated. A key enzyme for the biosynthesis of Arabidopsis major coumarin called scopoletin and its derivatives is Feruloyl-CoA 6′-Hydroxylase1 (F6′H1) that belongs to a large enzyme family of the 2-oxoglutarate and Fe(II)-dependent dioxygenases [3,5,10]. Our group elucidated the biological role of another member of this family, encoded by a strongly Fe-responsive gene (At3g12900), as a scopoletin 8-hydroxylase (S8H) involved in the last step of fraxetin biosynthesis [7]. Fraxetin is a coumarin derived from the scopoletin pathway, containing two adjacent hydroxyl groups in the ortho-position that can efficiently solubilise Fe from the hydroxide precipitates [3,8]. We proved S8H to be involved in coumarin biosynthesis as part of the Fe acquisition machinery [7].

Fe is a crucial micronutrient for every kind of living organism. It plays an essential role in metabolic processes such as DNA synthesis, respiration, photosynthesis, and it is a cofactor of many enzymes. The role of Fe homeostasis in resistance to infections was also shown across all kingdoms of life—different types of pathogens are likely to compete with their hosts for the acquisition of Fe [11]. Mechanisms of Fe homeostasis in plants, pathogens, and beneficial microorganisms play key roles in plant-microbe interactions [12]. Moreover, one-third of the world’s agricultural area is composed of calcareous soils, in which high pH leads to the precipitation of Fe that is finally not available and generate severe plant growth perturbation. Therefore, Fe deficiency is a widespread agricultural problem that reduces plant growth and crop yields, particularly in alkaline soils [13].

In addition to the important role in maintaining Fe homeostasis, coumarins can affect plant growth and fitness directly through their high biological activities including antibacterial and antifungal properties. Scopoletin that is one of the major Arabidopsis coumarins accumulating in roots [5,7,14], was shown to possess antimicrobial activity against various phytopathogens like *Ralstonia solanacearum* [15], *Alternaria alternata* [16], *Botrytis cinerea* [17], *Fusarium oxysporum* f. sp. *batatas* [18], *Sclerotinia sclerotiorum* [19], *Aspergillus flavus*, and *Aspergillus niger* [20], *Ceratocystis fimbriata* f. sp. *platani* [21] and acting against human pathogens [22] including multidrug-resistant *Pseudomonas aeruginosa* strains [23], *Salmonella typhi* [24] and clinical isolates of *Staphylococcus aureus* [25].

Recently, an important role for coumarins in microbiome modulation was demonstrated. It was shown that plant-derived coumarins shape the composition of Arabidopsis root microbial communities (rhizobiome) in Fe-starved plants, and possibly protect plants from pathogenic fungi [26]. Coumarins were shown to influence a reduced synthetic community (SynCom) of Arabidopsis root-isolated bacteria in synthetic media [27,28] and were proved to alter the rhizobiome and improve plant growth in Fe-limiting soil [29]. Even if, a role of root-exuded coumarins in structuring the rhizobiome was uncovered and their release into the rhizosphere upon pathogen infection was confirmed, the precise mechanisms underlying the above-described processes are only beginning to be discovered [26,28,30,31].

In the literature, there are examples of pathogens causing more severe disease symptoms on plants grown under high-Fe conditions when compared to Fe-deficient plants. One of the examples could be plant pathogenic bacteria *Dickeya dadantii* 3937, for which sufficient Fe uptake is essential to manifest full virulence on plants [32]. This bacterial pathogen causes soft rot and blackleg disease devastating potato and numerous other crops [33,34,35]. Bacteria from the *Dickeya* genus produce compounds called siderophores that form complexes with Fe and make it available to the microorganism. Taking into account that plants also produce Fe-chelating compounds, siderophore production is a part of the competitive relationships between plants and microorganisms that can promote infection [32]. It was demonstrated that Fe nutrition strongly affects the disease caused by *D. dadantii* 3937 also in a model plant Arabidopsis [36,37,38]. In Fe-starved Arabidopsis plants, authors observed a reduction in the expression of major bacterial virulence genes and finally a lower progression in disease symptoms on inoculated plants. The results obtained from a study of Arabidopsis response to *D. dadantii* 3937 infection highlight the major importance of the competition between plant and bacterial cells for Fe uptake during infection [36,38].

The above results reinforced the important role of coumarins in plant responses to disturbed Fe availability as well as their involvement in plant resistance and their release into the rhizosphere upon pathogen infection. The physiological functions of coumarins are strictly related to plant adaptation to various biotic and abiotic environmental stresses. Here, we investigated the Arabidopsis/*Dickeya* spp. pathosystem to better understand the relation between coumarins, plant Fe status, siderophores production and plant resistance to selected pathogenic bacteria.

## 2. Results

We used as a model, Arabidopsis wild-type plants (WT Col-0) and its mutants (*s8h* and *f6′h1*) defective in enzymes involved in coumarin biosynthesis (S8H and F6′H1, respectively) and *pdr9* mutant defective in coumarin transport to the rhizosphere (PDR9: Pleiotropic drug resistance 9 [2]). As plant pathogens, we included two *Dickeya* spp. strains: (1) a reference strain *D. dadantii* 3937 of medium virulence, (2) and *Dickeya solani* IFB0099 isolated from infected potato plant in Poland [39,40,41]. Interestingly, the selected strains differed in their ability to chelate Fe ions [40]. The ability to chelate Fe ions by *D. dadantii* 3937 strain was shown to be twice as high as the ability to chelate Fe ions by *D. solani* IFB0099 strain on CAS-agar medium [40]. 

### 2.1. Differential Susceptibility of Arabidopsis Plants Grown in Fe-Deficient Hydroponics to Tested Dickeya spp. Strains

To strictly control the growth conditions and maintain the nutrient composition of media, we conducted the hydroponic cultures (as described by [7] and [42]). Arabidopsis plants were cultivated in controlled conditions either under optimal (40 μM Fe^2+^) or Fe-deficient conditions (0 μM Fe^2+^) that induces coumarins accumulation [42] and subsequently were inoculated with *D. dadantii* 3937 and *D. solani* IFB0099 spp. It was reported previously that Arabidopsis roots release more phenolic-related compounds at later stages of life [43], therefore plants were inoculated at the flowering stage. We inoculated the Arabidopsis WT Col-0 and mutant lines defective in coumarin biosynthesis (*f6′h1*, *s8h*) with both *Dickeya* spp. strains, and evaluated disease progression according to the visual symptoms scoring with disease severity scale 0–5 (DSS) which allowed us to quantify the susceptibility of Arabidopsis plants to both bacterial strains (Figure 1).

We observed that independently of genetic characteristic, all Arabidopsis genotypes inoculated with *D. solani* IFB0099 strain developed more severe disease symptoms (DSS up to 4.5) (Figure 2a,b) when compared to *D. dadantii* 3937 (DSS up to 3.5) (Figure 2c,d). It is worth emphasizing that in case of all plant genotypes inoculated with *D. solani* IFB0099 strain, the symptoms of infection were more pronounced in conditions with optimal Fe availability (Figure 2a,b). Both mutants, in particular the *s8h* line, showed slightly higher disease symptoms after inoculation with *D. solani* IFB0099 when compared to WT Col-0 plants. Interestingly, when WT Col-0 plants were inoculated with *D. dadantii* 3937 strain, they developed the most severe disease symptoms at 96 h after inoculation when grown in optimal Fe solution (Figure 2c), while the opposite trend was detected for both Arabidopsis mutants with impaired biosynthesis of coumarins. During the *D. dadantii* 3937 infection, both Arabidopsis mutant lines showed a tendency to exhibit more severe disease symptoms in Fe-deficient hydroponics (Figure 2d) and not in the Fe-sufficient solution as WT Col-0 plants. This was particularly striking at 72 and 96 h after inoculation for the *f6′h1* mutants (Figure 2c,d), lacking the functional *F6′H1* gene, which does not synthesise scopoletin and its derivatives.

### 2.2. Inoculation of Arabidopsis WT Col-0 Grown in Fe-Deficient Hydroponics with *Dickeya* spp. Cause Decrease in the Expression of S8H and F6′H1 Genes Involved in Coumarin Biosynthesis

Next, to get insight into the expression levels of genes encoding key enzymes (S8H, F6′H1) involved in the biosynthesis of coumarins that are accumulated mostly in the underground part of a plant, we performed the qPCR analysis with cDNA reverse transcribed using RNA isolated from the WT Col-0 roots grown under different Fe-regimes and inoculated with *D. dadantii* 3937 or *D. solani* IFB0099 strains. As expected, we observed up-regulation of both genes (*S8H* and *F6′H1*) in the Fe-deficient condition (Figure 3a,b). In particular, the *S8H* gene, which is known to be one of the most strongly Fe-responsive genes, was induced several hundred times in all treatments tested in Fe-deficient conditions compared to Fe-sufficient condition (Figure 3a).

Interestingly, the expression of the *S8H* gene was significantly higher in the roots exposed only to abiotic stress (mock-inoculated WT Col-0 plants grown under Fe deficiency) when compared with those exposed to abiotic and biotic stress (*Dickeya* spp.- inoculated ones grown under Fe deficiency). It should be noted that the expression of *S8H* was relatively lower when combined environmental stress, composed of Fe-deficiency and bacterial infection, was applied (Figure 3a: right panel). This relative decrease was particularly significant when *D. dadantii* 3937 was used to inoculate WT Col-0 plants (*p* < 0.05). A similar trend was observed for the *F6′H1* gene, for which a relative reduction in the expression levels was observed when the same two stress factors, Fe-deficiency and bacterial inoculation, were simultaneously applied. The *F6′H1* expression was approximately 2-fold lower in WT Col-0 plants subjected to biotic stress when compared to plant exposed only to abiotic stress under Fe-deficient condition (Figure 3b). In this case, the inoculation with *D. dadantii* 3937 also had a significant effect on the relative reduction of *F6′H1* expression in the WT Col-0 genetic background (*p* < 0.05). That is an interesting observation considering that this strain expressed a greater ability than *D. solani* IFB0099 to chelate Fe ions, as shown previously by the CAS-agar plate assay [40].

### 2.3. Fe-Chelation in CAS Agar Plate Assay

To test the potential of plant-produced compounds to affect the bacterial siderophore production, we observed the growth and halo production of *D. dadantii* 3937 and *D. solani* IFB0099 strains on CAS-agar plates supplemented with homogenates prepared from the leaves of *s8h* and *f6′h1* mutants (Figure 4a,b) that were grown in Fe-depleted hydroponic solution (0 µM Fe^2+^).

Here, we selected leaves to prepare homogenates, as organs in which inoculation is conducted. No supplements or homogenates were added as a negative control (Ø), as a mock the phosphate buffer was used. It seems that both leaf homogenates and phosphate buffer itself can induce the production of bacterial siderophores. As could be expected, in most cases *D. dadantii* 3937 strain that was previously characterised by a higher ability to chelate Fe ions [40], produced larger halos compared to *D. solani* IFB0099 (Figure 4a,b). But the most interestingly, we observed the opposite effect when homogenates originating from the leaves of *f6′h1* mutant were used as a supplement. We observed a significant increase in *D. solani* IFB0099 halos’ diameter on agar plates supplemented with the *f6′h1* mutant leaf homogenates (Figure 4a,b).

### 2.4. Differential Susceptibility of Soil-Grown Arabidopsis Plants to Tested Dickeya spp. Strains

To shed light on the relationship between Fe homeostasis, coumarin accumulation and plant immunity in more physiological conditions, we grew a set of Arabidopsis mutants defective in coumarin accumulation (*f6′h1*, *s8h*) in the non-sterile soil environment. Here, we included in the experiment the *pdr9* mutant that is defective in coumarin transport to the rhizosphere [2]. Since soil mixture composition can alter Arabidopsis susceptibility to plant pathogens as was shown for the *Pseudomonas syringae* infection [45], we decided to estimate the disease symptoms caused by *D. dadantii* 3937 and *D. solani* IFB0099 on Arabidopsis plants grown in two different soil mixes (#1 and #2) derived from the commercial products that differ mainly in the level of salinity that can affect the availability of nutrients including Fe, and the content of some macro- and micronutrients like chlorides, phosphorus, potassium or calcium (Table 1).

Arabidopsis plants of all genotypes (WT Col-0, *f6′h1*, *s8h* and *pdr9*) had better performance on soil mix #1, on which the plant rosettes were much bigger. In case of soil mix #2, the spontaneous plant wilt and die was also observed. This is an interesting observation since the conducted chemical analysis did not show any dramatic differences in the compositions of both soil mixes (Table 1). However, regardless of the soil in which the plants were grown, the Arabidopsis of all tested genotypes inoculated with *D. solani* IFB0099 strain developed more severe infection symptoms (DSS up to 4.5), compared to those challenged with *D. dadantii* 3937 reference strain (DSS up to 2.5) (Figure 5). However, most interestingly, we detected a variation in the disease symptoms between plant genotypes inoculated with the *D. solani* IFB0099 strain. In soil mix #1 characterised by a slightly higher salinity, the *pdr9* mutants inoculated with *D. solani* IFB0099 showed the mildest infection symptoms among all plant genotypes with the severity score up to 3.5 (Figure 5d) compared with the DSS up to 4.0 for WT Col-0 and *f6′h1* plants (Figure 5a,c) and 4.5 for *s8h* mutant line (Figure 5b). In soil mix #2, a slightly lower DSS (lower than 3) was observed for all mutant lines inoculated with *D. solani* IFB0099 (Figure 5b–d) compared to WT Col-0 plants (DSS up to 3) (Figure 5a). We did not observe such a variation in the infection symptoms on plants inoculated with the *D. dadantii* 3937 strain.

### 2.5. Expression of Selected Plant Stress-Response Genes Is Strongly Induced in Arabidopsis Mutants Defective in Coumarin Accumulation Inoculated with D. solani IFB0099

Next, to measure the expression levels of selected genes being the plant stress markers, we analysed by qPCR the expression of genes related to oxidative stress (*SOD1*, AT1G08830), plant defence (pathogenesis-related *PR1*, At2g14610) and modulation of jasmonate-induced root growth inhibition (*CYP82C2*, At4g31970). We used RNA isolated from the leaves of Arabidopsis WT Col-0 and three mutant lines (*f6′h1*, *s8h*, *pdr9*) grown in soil mix #1, in which the disease symptoms after *D. solani* IFB0099 or *D. dadantii* 3937 strains infection were more prominent compared to soil mix #2 (Figure 5). 

We observed a strong increase in expression of selected plant stress markers in plants inoculated with *D. solani* IFB0099 strain, which indicate that inoculation of Arabidopsis plants with this pathogen particularly induce the plant defence systems (Figure 6a–c). Interestingly, the expression levels of two out of three tested genes encoding plant stress markers (SOD1, PR1) were differentially induced among mutants and WT Col-0 plants, those inoculated with *Dickeya* spp. strains and mock-inoculated ones (Figure 6a,b). The level of the *SOD1* gene was visibly higher in all mutant lines compared to WT Col-0 plants (Figure 6a). While in the case of the *PR1* gene, we observed the opposite effect, where its expression was higher in WT Col-0 plants compared to mutants with disturbed coumarin biosynthesis (*f6′h1*, *s8h*) or transport (*pdr9*) (Figure 6b). The transcript levels of *CYP82C2* were specifically induced by *D. solani* IFB0099 infection in all tested plant genotypes (Figure 6c).

## 3. Discussion

Microorganisms that urgently need Fe for their growth, replication, metabolism and the infectious disease process, have evolved numerous strategies for Fe acquisition such as siderophore production. At the same time, plants are constantly subjected to various environmental stresses, including Fe-deficiency or pathogen attack during which Fe itself plays an important role. During a microbial infection, there is a competition between host and pathogen for the necessary nutritional resources. Numerous studies have shown that Fe ions play a key role in such competitive relationships [12,46,47,48,49]. It was also shown recently by several groups, including our research team, that the secretion of coumarins is essential for Fe acquisition under Fe-deficient conditions in a model plant *Arabidopsis thaliana* [7]. Plants and microorganisms have evolved a set of active strategies for Fe uptake from the soil that are based on acidification, chelation and reduction processes. Root exudation is one of such important processes determining the interaction of plants with the soil environment and microbiome. Coumarins that are secreted to the rhizosphere by roots are involved in several processes determining plant interactions with the soil environment, both with biotic and abiotic factors.

In this study, we evaluated the disease symptoms caused by *Dickeya* spp. strains in Arabidopsis lines differing in coumarin accumulation that were grown under various growth conditions and Fe availability. The use of selected Arabidopsis mutants and bacterial strains of different origin enabled us to compare [1] the variation in disease symptoms among plant genotypes under numerous environmental scenarios, [2] and the expression of stress-related genes in plant genetic backgrounds with disturbed production and distribution of coumarins.

The presented analyses provided interesting insights into the differences in responses of the following plant genotypes: control plants (WT Col-0), coumarin-reach plants (WT Col-0 growing in Fe deficient environment), coumarin-deficient plants (*f6′h1*), fraxetin-deficient plants (*s8h*) and coumarin-hyperaccumulating plants (*pdr9*) to *D. dadantii* 3937 and *D. solani* IFB0099 strains. The developed model system of Arabidopsis/*Dickeya* spp. was applied to investigate the effect of two abiotic factors (Fe availability and coumarin content) on the disease severity. These studies are in line with the latest Top 10 Questions, selected by the International Congress on Molecular Plant-Microbe Interactions (IC-MPMI) community that met in Glasgow in 2019, covering the need to understand how the abiotic environment influence specific plant-microbe interactions [50]. It should be also highlighted that most of the previously published data describe the interaction between Arabidopsis plants and *D. dadantii* 3937. According to our best knowledge *D. solani*, which is an important plant pathogenic bacterium causing a loss in potato yield in Europe [34,51,52] was not tested before with a model plant Arabidopsis. Moreover, recent results show that *D. solani* strains cause severe disease symptoms in temperate climates, and are more aggressive than other blackleg-causing bacteria from genus *Dickeya* and *Pectobacterium* spp. [33,35,53,54].

To shed light on the strong relationship between Fe homeostasis (abiotic factor), coumarin accumulation and plant susceptibility to plant pathogenic bacteria, we grew WT Col-0 plants and two Arabidopsis mutants defective in coumarin accumulation (*s8h* and *f6′h1*) in the hydroponic cultures with strictly controlled Fe content and inoculated them with *Dickeya* spp. strains (biotic stress factor). We observed that all tested Arabidopsis genotypes (WT Col-0, *s8h* and *f6′h1*) inoculated with *D. solani* IFB0099 strain developed more severe disease symptoms than plants inoculated with *D. dadantii* 3937. The disease symptoms associated with *D. solani* IFB0099 infection were much more pronounced in Fe-sufficient hydroponics. A similar effect was observed for *D. dadantii* 3937-inoculated wild-type plants (WT Col-0). This is in line with the literature data showing that *D. dadantii* belongs to the pathogens causing more severe disease symptoms on plants grown under high-Fe conditions when compared to Fe-deficient environmental condition [32]. The most noticeable was a detection of the opposite effect for Arabidopsis mutants with impaired biosynthesis of coumarins. During *D. dadantii* 3937 infection of *s8h* and *f6′h1* mutant plants, the more severe disease symptoms were observed in Fe-deficient hydroponics, and not in the Fe-sufficient cultures, particularly at 72 h after inoculation. Both of these mutants (*s8h* and *f6′h1*) are defective in enzymes involved in the biosynthesis of coumarins, which are secondary metabolites important for Fe uptake in plants [4]. Consequently, the *Dickeya* spp. cells, which compete for Fe with the plant cells, can uptake Fe with higher efficiency and accumulate more of it than those infecting WT plants. It was shown before by other groups [32,37] that Fe uptake is important for bacteria ability to macerate plant tissue and the production of virulence factors. As a result of this, the *Dickeya* spp. strains infecting Arabidopsis mutants, which are defective in Fe acquisition, cause more severe disease symptoms in these plants. It is worth noticing that qPCR analysis of the corresponding genes (*S8H* and *F6′H1*) in the WT Col-0 genetic background, proved that their expression levels were relatively lower when a combined environmental stress, composed of Fe-deficiency and bacterial infection, was applied. Taking into account that *D. dadantii* 3937 has a higher ability to chelate Fe ions and the expression of several bacterial genes involved in siderophore-mediated Fe uptake is controlled by the Fe availability [32], it can be concluded that (1) coumarins produced by plants influence more strongly pathogens for which siderophores production play a particularly important role in the pathogenesis process, (2) disorders of coumarin biosynthesis are more important for the disease symptoms under Fe-deficiency conditions.

Next, we explored the potential of coumarins and other factors possibly present in the selected plant homogenates to affect the bacterial siderophore production by measuring the halos’ diameter produced by the *Dickeya* spp. strains on CAS-agar plates. We tested leaf homogenates prepared from a set of Arabidopsis plants (WT Col-0, *s8h* and *f6′h1*). In siderophore production assay, we observed greater halos produced by *D. dadantii* 3937 than that produced by *D. solani* IFB0099 for all tested Arabidopsis genotypes, except for the *f6′h1* mutants. It seems like coumarins and other possible factors present in leaf homogenates and phosphate buffer can induce, directly or indirectly, the production of bacterial siderophores. The existence of not-characterised yet interplay between coumarins and the bacterial siderophore production needs further investigation. It is important to continue and develop research on the role of coumarins as novel elements of chemical communication and to test if, in coumarin-deficient plants, the induction of other compensatory pathway occurs.

To better understand the responses of coumarin-deficient plants to combined environmental stimuli, we grew a set of Arabidopsis mutants defective in coumarin accumulation (*f6′h1*, *s8h*) and coumarin transport to the rhizosphere (*pdr9*) in more physiological conditions. The inoculation was conducted on plants grown in two soil mixes with some differences in chemical compositions (Table 1). It has to be highlighted that plants grew significantly better in soil mix #1, in which all rosettes were much larger. We can speculate that the lack of any of the elements of soil mix #2 is limiting plant growth. However, considering that the soil mix #2 consist of half of the peat moss, which is a natural product of organic origin, we can suspect the significant differences in the microbiomes of the tested soil mixes. These interesting questions should be clarified in the future. Importantly, regardless of the soil mix in which plants were grown, the Arabidopsis of all tested genotypes inoculated with the *D. solani* IFB0099 strain developed more severe infection symptoms compared to *D. dadantii* 3937 reference strain. For both bacterial strains, the symptoms of infection were more pronounced on plants with a better growth on soil mix #1 compared to plants grown in soil mix #2. Interestingly, we detected a variation in the disease symptoms between plant genotypes inoculated with *D. solani* IFB0099 strain, particularly on plants grown in the soil mix #1. The *pdr9* mutant plants that hyperaccumulate coumarins in their tissues, showed the mildest infection symptoms among all plant genotypes when inoculated with *D. solani* IFB0099 strain. While both Arabidopsis mutants (*f6′h1* and *s8h*) defective in coumarin biosynthetic genes, showed stronger symptoms of infection. The explanation of this phenomenon can be that coumarins are known for their antimicrobial activity, however, the observed genotype-specific mode of action needs further investigations. We did not observe such a clear variation in the infection symptoms on plants inoculated with *D. dadantii* 3937 strain that is characterised by a higher ability to chelate Fe ions (the CAS-plate assays presented in Figure 4 and [40]).

In this work, we also analysed the expression profiles of three plant genes (*PR1, SOD1, CYP82C2*) which products are involved in the plant tissue response to a wide range of stresses including biotic and abiotic factors. During *D. solani* IFB0099 infection of Arabidopsis, the expression of *PR1*, which is considered to be one of the markers for salicylic acid (SA)-dependent systemic acquired resistance (SAR) [55,56], was strongly induced in the leaves of all infected genotypes. This increase in the *PR1* gene expression was most pronounced in the WT Col-0 genetic background. A similar *PR1* expression profile was observed in *D. dadantii* 3937- and mock-inoculated plants, but the levels of *PR1* expression were much lower in these experimental setups. In parallel, we observed in *D. solani* IFB0099-inoculated plants an induction in the expression of the *SOD1* gene, which encodes a cytosolic copper/zinc superoxide dismutase CSD1 that can be regulated by biotic and abiotic stresses and detoxify superoxide radicals [57]. This indicates that plants infected with *D. solani* IFB0099 induce the defensive mechanism by increasing the production of critical antioxidant enzymes protecting organisms from reactive oxygen species. Interestingly, plants with impaired biosynthesis or coumarin accumulation have a higher expression of *SOD1*, although the observed differences are not statistically significant. Furthermore, the expression of *CYP82C2* in *D. solani* IFB0099-inoculated Arabidopsis WT Col-0 and its mutants, was also clearly upregulated in comparison to *D. dadantii* 3937- and mock-inoculated plants. Since *CYP82C2* was shown to be involved in several aspects of jasmonic acid (JA) responses [58], it seems likely that infection with *D. solani* IFB0099 pathogen induces activation of the JA-dependent response.

The above results and data previously published obtained from a study of Arabidopsis response to *D. dadantii* 3937 infection highlight the major importance of the competition between plant and bacterial cells for Fe uptake during infection [36,38]. It was demonstrated that Fe nutrition strongly affects the disease caused by soft rot-causing plant pathogenic bacteria with a large plant host range including Arabidopsis. Plants have evolved various strategies to acquire Fe from their environment and mechanisms tightly regulating Fe uptake, transport and storage [11,13,59] including the production of Fe-mobilizing phenolic compounds like coumarins [1,2,3,4,5,7,8,60,61,62]. The production of exudates, which is dependent on the external environment, at the same time is genetically regulated in plants. It was shown by Micallef et al. [63] that natural populations of Arabidopsis originating from various geographical localization, called accessions, release a unique set of compounds into their rhizosphere. The authors detected that the rhizobacterial community composition and the relative abundance of particular ribotypes were also accession-dependent. They hypothesised that the observed natural variation in root exudation could partly explain the genotypic influence on bacterial communities in the rhizosphere [63]. Many studies of plant-microbe interactions revealed that plants are not only able to shape their rhizosphere microbiome, but also highlight this root-associated microbial community to be referred to as the second genome of the plant, which is crucial for plant health [64]. Our research group detected previously the existence of natural variation in the accumulation of antimicrobial coumarins, namely scopoletin and scopolin, among Arabidopsis accessions [14]. Lately, we also detected a significant variation in the content of other simple coumarins like umbelliferone and esculetin together with their glycosides: skimmin and esculin, respectively [65]. It was also shown recently that a natural variation exists in Arabidopsis tolerance to *Dickeya* spp. [66]. The significantly different susceptibility groups were uncovered within a small set of eight Arabidopsis accessions following inoculation with *D. dadantii* 3937, which suggested that tolerance associated loci might be present in this model plant. Even though *Dickeya* spp. are causative agents of severe diseases in a wide range of plant species and major economic losses, little data concerning potential resistance genes are available [67,68,69]. These data strongly suggest that Arabidopsis with its extensive genetic natural variation and a set of powerful genetic tools including web-accessible collections of mutants, provides an excellent model to study the interplay between secondary metabolites production, exudate profiles, Fe homeostasis and interaction with beneficial and plant pathogenic microbes.

## 4. Materials and Methods

### 4.1. Plant Material

The Arabidopsis thaliana accession Columbia was used as the wild type (Arabidopsis WT Col-0) together with a set of T-DNA insertional mutant lines in the Col-0 background: [1] *s8h-1* (SM_3.27151); *s8h-2* (SM_3.23443); [2] *f6′h1-1* (SALK_132418); *f6′h1-2* (SALK_050137C) and [3] *pdr9-1* (SALK_050885). Seeds of all lines are available at the Nottingham Arabidopsis Stock Centre (http://arabidopsis.info/, accessed on 15 April 2021).

### 4.2. Bacterial Strains, Media, Growth Conditions

The strains used in this study, *Dickeya dadantii* 3937 IFB0459 and *Dickeya solani* IFB0099, are available at the collection of bacterial pathogens located at the IFB UG & MUG, in Poland. For the plant inoculation, the bacteria were grown overnight in lysogeny broth (LB) [70] liquid medium at 28 °C with agitation at 120 rpm. Then the bacterial cultures were centrifuged in Eppendorf tubes (5 min, 6500 rpm), washed in sterile 0.85% NaCl and centrifuged again. The bacterial suspension (at least 15 mL) was prepared in sterile 0.85% NaCl and adjusted to 1 MacFarland Unit (Densitometer DEN-1/DEN-1B, Buch & Holm Herlev, Denmark), approximately 10^8^ cfu/mL.

### 4.3. Hydroponic Cultures

After a few days’ stratification at 4 °C, Arabidopsis WT Col-0 plants were grown in a controlled environment (16 h light at 22 °C/~100 μmol m^−2^ s^−1^ and 8 h dark at 20 °C) in a modified 1 × Heeg solution [71], as described in details in [42] with the following modifications. Approximately 3-weeks-old plants in tube lids, filled with the solidified Heeg medium, were transferred from tip boxes with control solution (40 µM Fe^2+^) to the modified 50 mL Falcon centrifuge tubes filled with optimal (40 μM Fe^2+^) or Fe-deficient (0 μM Fe^2+^) medium. The roots were passing through the 1-cm diameter hole drilled in Falcons’ lid to support the seedling holder as proposed by Conn et al. [72]. Hydroponic solutions were replenished by the addition of a fresh medium.

### 4.4. Soil Cultures

Arabidopsis seeds were first stratified in Petri dishes on water-saturated Whatman paper followed by a cold treatment for 4 d at 4 °C and then planted into two different soil mixes derived from commercial products. Soil mix #1 consists of commercial soil 1 and vermiculite (3–6 mm in diameter) in a proportion of 3:1, respectively. Soil mix #2 consists of commercial de-acidified peat moss mix with commercial soil 2 and vermiculite in a proportion 2:1:1, respectively. Chemical analysis were conducted by the Regional Agro-Chemical Station in Gdansk, Poland (OSCh-R, http://www.oschrgdansk.pl/, accessed on 15 April 2021). Prior to sowing seeds, both soil mixes were soaked with general-purpose fertiliser (Substral, The Scotts Miracle-Gro Company, Marysville, OH USA). Arabidopsis plants were grown for five weeks under a photoperiod of 16 h light (120 μmol m^−2^ s^−1^) at 22 °C and 8 h dark at 20 °C, before being inoculated.

### 4.5. Siderophore Production in the Presence of Plant Extracts

Homogenates from leaves of *Arabidopsis thaliana* were prepared from WT Col-0 plants, *s8h* and *f6′h1* mutants. Briefly, the pooled leaves (~300 ± 40 mg) stored at −80 °C were thawed and homogenised in Bioreba bags (BIOREBA AG, Reinach, Switzerland) with 3 mL of 50 mM phosphate buffer pH 7.0 with the use of hand homogeniser (BIOREBA AG, Reinach, Switzerland). Then, the homogenate was centrifuged twice in Eppendorf tubes (8500 rpm, 2 min). The supernatant was filtered with a syringe 0.22 µm filter into sterile Eppendorf tubes (in total 3.5 mL) and immediately used on plates. Siderophore production of bacterial strains was determined on chrome azurol S–agar (CAS-agar) plates [73] supplemented with 100 µL of each homogenate with sterile spreader 15 min before bacteria inoculation. The overnight cultures of bacterial strains were centrifuged and resuspended in sterile 0.85% NaCl and adjusted to 0.5 MacFarland (Densitometer DEN-1/DEN-1B, Buch & Holm). 10 µL of each bacterial suspension was put on the CAS-agar plates and incubated at 28 °C for up to 168 h. We measured the halo diameters developed on CAS-agar plates supplemented with plant homogenates every 24 h.

### 4.6. Plant Inoculation with Bacterial Strains

Approximately 5-weeks-old plants, grown in soil or hydroponically, were inoculated with the bacterial suspensions of either *D. dadanti* 3937 or *D. solani* IFB0099 with the use of laboratory pincers. The pincers were sterilised before use and approximately 1 cm of the pincer tip was dipped into the bacterial suspension (the final inoculum of 10^8^ cfu/mL) and immediately the plant leaf was pinched with the pincers. At least 8 leaves were inoculated with each bacterial strain and mock. We pinched the middle parts of the selected representative leaves (3 leaves per plant). The negative control (mock) were plants inoculated only with sterile 0.85% NaCl. For the mock-inoculated plants no symptoms development was observed throughout the experiment. The number of bacteria inoculated into the plant leaf with pincers was about 2 × 10^7^ cfu/leaf and it was stable throughout experiments (data not shown). After inoculation, trays with plants were placed in the boxes filled with one litre of water, which were covered with transparent lids, to enable 100% humidity. Next, boxes were placed for 96 h in phytotron at 28 °C (16 h light at 28 °C/~100 μmol m^−2^ s^−1^ and 8 h dark at 26 °C), which is an optimal temperature for the development of disease symptoms by bacteria from the *Dickeya* spp. The whole rosettes were collected for each genotype grown in soil mix #1, frozen in liquid nitrogen and stored at −80 °C until further analysis. The roots of Col-0 plants grown hydroponically were gently removed from the agar droplets with tweezers and then rinsed in a beaker with distilled water. After drying on a paper towel, roots were frozen in liquid nitrogen and placed at −80 °C.

### 4.7. Quantification of Arabidopsis Plants Susceptibility to *Dickeya* spp. Strains by Visual Symptom Scoring (Disease Severity Scale, DSS)

The Arabidopsis plants were scored for soft rot/ blackleg symptoms development on leaves daily at 0 h, 24 h, 48 h, 72 h and 96 h post-inoculation. Developed symptoms of the disease were assigned to 0–5 scale (Figure 1), for which “0” meant no signs of symptoms of the disease and the wound has healed (it was observed for the negative control leaves); “1” the necrotic tissue was observed in the inoculation site only; “2” the necrotic tissue observed in the inoculation site and max. 3 mm wide around it; “3” the maceration visible around the inoculation site spreading further with possible chlorosis of the leaf; “4” visible maceration of the whole leaf, possible chlorosis of other leaves, no maceration of the limb; “5” visible maceration of the whole limb and the leaf.

### 4.8. RNA Extraction and Expression Analysis by qRT-PCR

Total RNA was extracted from plant material harvested 48 h after inoculation. The plant tissue was homogenised in liquid nitrogen using sterile mortars cleaned with isopropanol and baked for 4 h at 180 °C. A commercially available E.Z.N.A.^®^ Plant RNA Kit (Omega Bio-tek, Inc., Norcross, GA USA) was used following the instructions of the manufacturer and including an additional step to remove the genomic DNA contamination from the mixture with RNase-Free DNase I Set (Omega Bio-tek, Inc., Norcross, GA USA). 500 ng RNA for RNA isolated from leaves or 200 ng RNA for RNA isolated from roots was used for reverse transcription by Maxima First Strand reverse transcriptase cDNA Synthesis Kit for RT-qPCR (Thermo Fisher Scientific, Waltham, MA USA). qPCR was performed using LightCycler^®^ 480 Real-Time PCR System (Hoffmann-La Roche, Basel, Switzerland) and SYBR^®^ Green Master Mix (Thermo Fisher Scientific, Waltham, MA USA), using the gene-specific primers shown in Table 2. Primers’ specificities were confirmed by the analysis of the melting curves. Relative transcript levels (RLT) of the plant genes in leaf tissues were normalised to the transcript level of the house-keeping *ACTIN2* gene (At3g18780). As a reference for the root tissues, the *EF-1α* (ang. elongation factor-1α, At5g60390) gene was selected [44].

## 5. Conclusions

We investigated here the possible interactions between plant resistance, coumarin content and Fe status by using the plant pathogenic *Dickeya* spp. strains and a set of selected Arabidopsis mutants defective in coumarin biosynthesis (*f6′h1*, *s8h*) and their transportation (*pdr9*). We studied the effect of disturbed coumarin accumulation and Fe deficiency on the disease severity using a model system of Arabidopsis/*Dickeya* spp. interactions. Arabidopsis plants grown in hydroponic cultures with different Fe regimes and two soil mixes were inoculated with *Dickeya* spp. strains or treated with NaCl as a control. Under all conditions tested, Arabidopsis plants inoculated with *D. solani* IFB0099 developed more severe disease symptoms compared to plants inoculated with *D. dadantii* 3937 strain. While the response of plants to *D. dadantii 3937* infection was genotype-dependent in Fe-deficient hydroponic solution. Subsequently, we showed that the expression of genes encoding plant stress markers was also strongly induced by *D. solani* IFB0099 infection. Interestingly, the inoculation of WT Col-0 plants grown in Fe-deficient hydroponics with both *Dickeya* spp. strains cause a decrease in the expression of *S8H* and *F6′H1* genes involved in coumarin biosynthesis. 

*Dickeya* spp. was chosen as a plant pathogenic bacteria causing soft rot disease that can infect a broad spectrum of plants, whereas plant genotypes were selected due to their disturbed coumarin accumulation in roots and exudate profiles, which may have an impact on specific microbial consortia selection in the rhizosphere and influence plant response to pathogen attack. This may play a particularly important role for plant development and growth under Fe deficiency. The molecular mechanisms underlying these fascinating interactions are not yet well understood. We believe that Arabidopsis*/Dickeya* spp./pathosystem, together with a set of various Arabidopsis mutants defective in coumarin biosynthesis and its significant natural genetic variation, will be in future beneficial in uncovering a role of root-exuded coumarins in structuring the rhizobiome and plant resistance to pathogens.

## Figures and Tables

**Figure 1 ijms-22-06449-f001:**
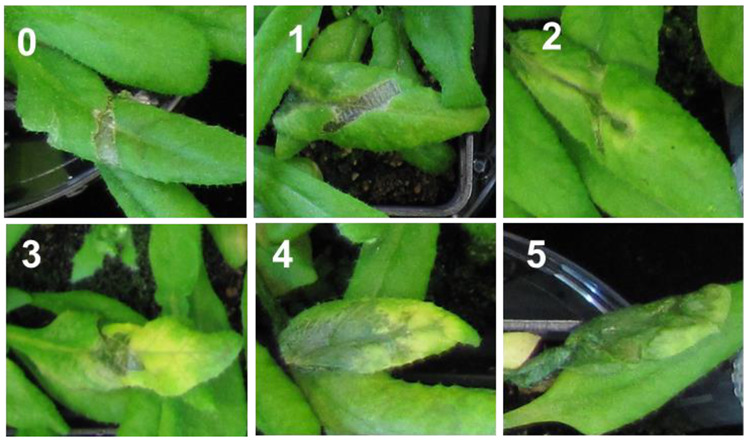
Disease severity scale (DSS) on *Arabidopsis thaliana* wild-type (WT) Col-0 leaves inoculated with *D. dadantii* 3937 and/or *D. solani* IFB0099. Example images representing stages of DSS were taken 48 h after inoculation. The DSS was assigned to 0–5 scale and was defined as: (**0**) for no signs of symptoms of the disease and the wound has healed (observed for the mock control); (**1**) the necrotic tissue was observed in the inoculation site only; (**2**) the necrotic tissue observed in the inoculation site and max. 3 mm wide around it; (**3**) the maceration visible around the inoculation site spreading further with possible chlorosis of the leaf; (**4**) visible maceration of the whole leaf, possible chlorosis of other leaves, no maceration of the limb; (**5**) visible maceration of the whole limb and the leaf.

**Figure 2 ijms-22-06449-f002:**
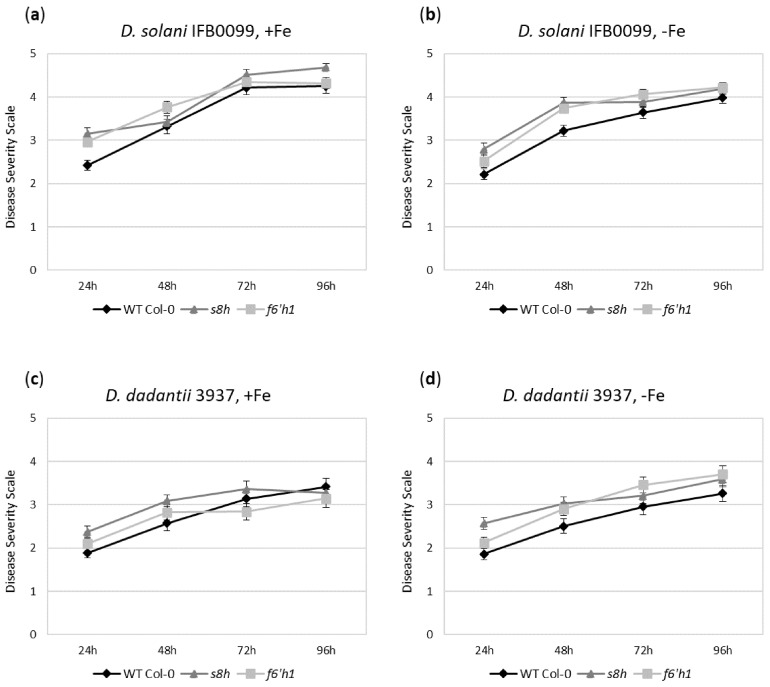
Disease progression caused by *Dicekya solani* IFB0099 (**a**,**b**) and *Dicekya dadantii* 3937 (**c**,**d**) strains on *Arabidopsis thaliana* wild-type (WT) Col-0 plants, *s8h* and *f6′h1* mutant lines grown in optimal (+Fe; 40 µM Fe^2+^) (**a**,**c**) or Fe-deficient (−Fe, 0 µM Fe^2+^) (**b**,**d**) hydroponic cultures by visual symptom scoring (Disease Severity Scale, DSS). The values represent the mean values of DSS originating from two independent experiments, in each experiment numerous individuals (*n* = 5–9) per plant genotypes (three leaves per plant) were inoculated for each time point. It is worth noting that the results averaged the DSS values obtained for two independent mutant lines for each tested gene. The mock-inoculated plants (with 0.85% NaCl) did not show the symptoms of the disease progression throughout the experiment. Error bars represent ± standard error (SE).

**Figure 3 ijms-22-06449-f003:**
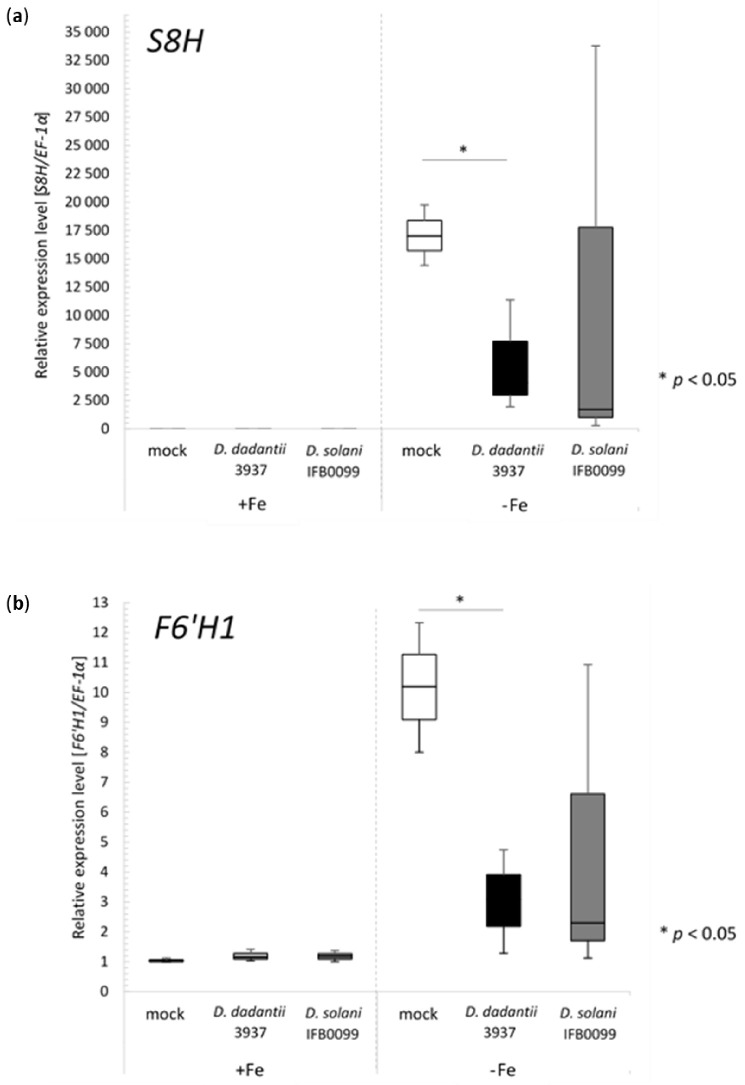
The relative expression levels of *S8H* (**a**) and *F6′H1* (**b**) genes were analysed in the roots of *Arabidopsis thaliana* wild-type (WT) Col-0 grown in hydroponics under optimal (40 μM Fe^2+^) or Fe-deficient conditions (0 μM Fe^2+^) and inoculated with *Dickeya dadantii* 3937 and *Dickeya solani* IFB0099 strains. Control plants were mock-inoculated with a 0.85% NaCl solution. As a reference, the *EF-1α* (ang. elongation factor-1α, At5g60390) gene was selected [44]. The pairwise t-test was used. Error bars, ±SD, from three biological replicates. * *p* < 0.05.

**Figure 4 ijms-22-06449-f004:**
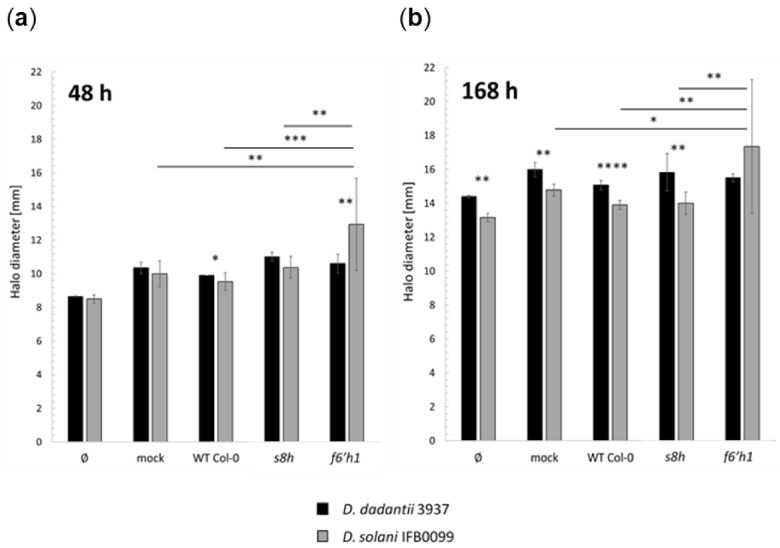
Siderophore production by *Dickeya dadantii* 3937 and *Dickeya solani* IFB0099 after incubation for (**a**) 48 and (**b**) 168 h on CAS-agar plates supplemented with leaf homogenates of *Arabidopsis thaliana* wild-type (WT) Col-0 and *s8h* or *f6′h1* mutant plants. The halo diameters were measured [mm] on control CAS-agar medium with no supplements (Ø), mock (phosphate buffer) and CAS-agar medium supplemented with leaf homogenates prepared from the Col-0 WT plants or *f6′h1, s8h* mutants. The experiment was performed twice, with 12 replicates (a set of pooled 3 plants was used for the biological replicates). The results are presented as average halo diameter. Error bars represent ± standard deviation (SD). Statistical significance is marked with asterisks * *p* < 0.05, ** *p* < 0.01, *** *p* < 0.001 and **** *p* < 0.0001 based on student’s *T*-test for independent samples.

**Figure 5 ijms-22-06449-f005:**
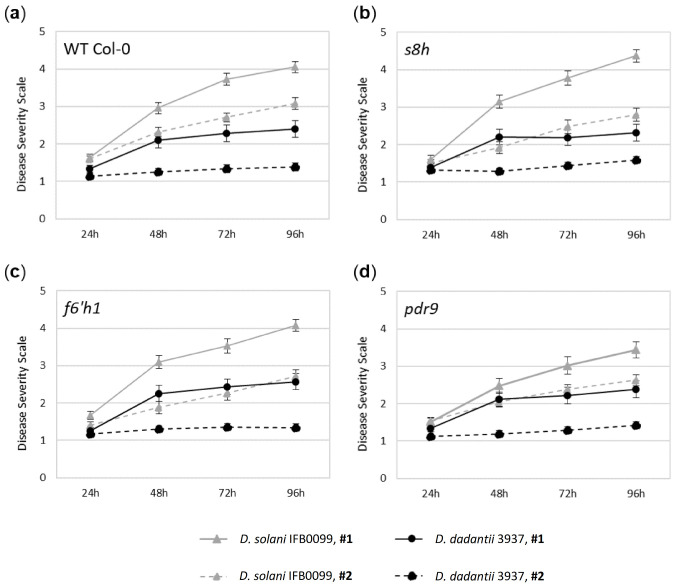
Disease progression caused by *Dickeya dadantii* 3937 and *Dickeya solani* IFB0099 strains on *Arabidopsis thaliana* (**a**) wild-type (WT) Col-0 plants, (**b**) *s8h*, (**c)** *f6′h1* and (**d**) *pdr9* mutant lines by visual symptom scoring (Disease Severity Scale, DSS). Plants were grown in two types of soil mixes (#1 and #2, see Table 1). The values represent the mean values of DSS originating from two independent experiments, in each experiment numerous individuals (*n* = 6–8) per plant genotypes (three leaves per plant) were inoculated for each time point. The mock-inoculated plants (with 0.85 % NaCl) did not show the symptoms of the disease progression throughout the experiment. Error bars represent ± standard error (SE).

**Figure 6 ijms-22-06449-f006:**
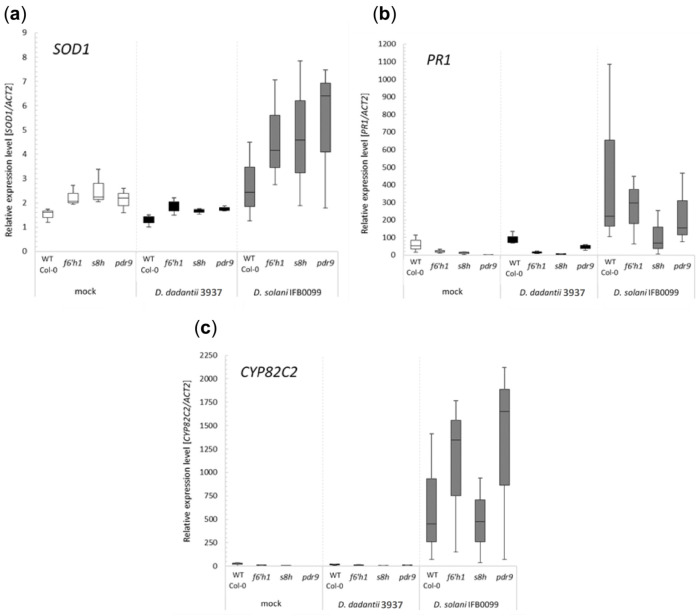
Relative expression levels of plant stress marker genes (**a**) *SOD1* (AT1G08830), (**b**) *PR1* (At2g14610) and (**c**) *CYP82C2* (At4g31970) analysed in the leaves of *Arabidopsis thaliana* wild-type (WT) Col-0 and mutant plants (*f6′h1, s8h, pdr9*) grown in soil mix #1 and inoculated with *Dickeya dadantii* 3937 and *Dickeya solani* IFB0099. 0.85% NaCl treated plants were used as a negative control. As a reference, the *ACT2* (At3g18780) gene was used [44]. Error bars, ±SD, from three biological replicates.

**Table 1 ijms-22-06449-t001:** Chemical analysis of used soil mixes: (**a**) pH, salinity, macro- and (**b**) micronutrients.

(**a**)								
**Soil Mix** **no.**	**pH in H_2_O**	**NaCl g/dm^3^ Soil**	**NO_3_**	**Cl**	**P**	**K**	**Ca**	**Mg**
			**mg/dm^3^ Soil**					
**#1**	6.8	2.65	224	13.6	34.6	70.1	2960	>400 (548) ^1^
**#2**	6.9	1.59	220	10.5	44.5	91.8	2509	>400 (498) ^1^
(**b**)								
**Soil Mix** **no.**	**Cu**	**Zn**	**Mn**	**Fe**	**B**			
	**mg/dm^3^ Soil**							
**#1**	1.1	1.1	1.3	51.4	0.4			
**#2**	1.0	1.3	2.3	49.2	0.4			

^1^ Results above upper limit of the method range for Mg = 400 mg/dm^3^.

**Table 2 ijms-22-06449-t002:** Primer sequences for plant genes used in qPCR reactions.

Name	Sequence (5′-3′)	Description
3g18780For	CTTGCACCAAGCAGCATGAA	Primer for *ACT2* gene ^1^
3g18780Rev	CCGATCCAGACACTGTACTTCCTT	Primer for *ACT2* gene ^1^
AT2G14610_F	TTCTTCCCTCGAAAGCTCAAGA	Primer for *PR1* gene
AT2G14610_R	GTGCCTGGTTGTGAACCCTTA	Primer for *PR1* gene
AT4G31970_F	GATGGTGAGAATGGTGGCCG	Primer for *CYP82C2* gene
AT4G31970_R	GCCTCTTCGGCATCTTCAGG	Primer for *CYP82C2* gene
AT1G08830_F	TCAACCCCGATGGTAAAACAC	Primer for *SOD1* gene
AT1G08830_R	TCACCAGCATGTCGATTAGCA	Primer for *SOD1* gene
At5g60390_F	TGAGCACGCTCTTCTTGCTTTCA	Primer for *EF-1α* gene ^1^
At5g60390_R	GGTGGTGGCATCCATCTTGTTACA	Primer for *EF-1α* gene ^1^
S8HqPCR_F	GCCGAGACACTTGGCTTCTT	Primer for *S8H* gene
S8HqPCR_R	CAGCAGCTCCACCGAAACA	Primer for *S8H* gene
F6H1qPCRf	TGATGAGGACAGAGTCGCTGAA	Primer for *F6′H1* gene
F6H1qPCRr	CACTTGAAAGAACCCCCATTTC	Primer for *F6′H1* gene

^1^ Reference [44].

## Data Availability

The data presented in this study are available within the article.

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
