# Peer review of "Interplay between Coumarin Accumulation, Iron Deficiency and Plant Resistance to Dickeya spp."

_ijms, 2021, doi:10.3390/ijms22126449_

Round 1
Reviewer 1 Report
The manuscript by Izabela Perkowska and colleagues aims at investigating on the role coumarin accumulation on Arabidopsis resistance to D. dadantii 3937 and D. solani IFB0099 strains. To do this, authors analyzed control plants (Col-0), coumarin-reach plants (Col-0 growing in Fe deficient environment), coumarin-deficient plants (f6’h1), fraxetin-deficient plants (s8h) and coumarin-hyperaccumulating plants (pdr9) under various growth conditions and Fe availability and found that disorders of coumarin biosynthesis are more important for the disease symptoms under Fe-deficiency conditions.
Overall, I find that the paper describes well the results of the experiments which are appropriate to reach the conclusions claimed by authors. The main weak point I see is related to the originality/novelty of the research. Nevertheless, I have few minor concerns that I think the authors should address before publication:
Figure 2: I wonder why coumarin-hyperaccumulating plants (pdr9) were not included and/or shown as a control in this experiment, although in the above line is listed among lines considered for the experiments?
Figure 2: I make some difficulty in comparing the behavior of the mutant lines with the wild type one. I wonder whether it would be better to combine graphs including the different Arabidopsis mutant and wt lines together in one graph and their responses to different pathogen strains in separate graphs?
Line 291: “Interestingly, the level of PR1 gene was specifically induced in WT Col-0 plants after mechanical wounding (see mock panel in Figure 6b). This was not observed for the mutant lines”.
Based on the analyses of what control/parameter authors conclude on the induction of PR1 gene expression in wounded tissues?
I guess an untreated/unwounded control should be considered to ascertain the grade of induction in wounded and mock treated tissues. Unless I missed it, such a control is not shown nor discussed.
Reviewer 2 Report
REVIEW REPORT
The manuscript # ijms-1205101 entitled “Interplay between coumarin accumulation, iron deficiency and plant resistance to Dickeya spp.” reports the effect of coumarin on iron deficiency and pathogen resistance in Arabidopsis/Dickeya spp. The authors found that Arabidopsis plants inoculated with Dickeya solani IFB0099 strain developed more severe disease symptoms compared to lines inoculated with Dickeya dadantii 3937. Furthermore, they showed that the expression of genes encoding plant stress markers (SOD1, pathogenesis-related PR1 and CYP82C2) were strongly affected by D. solani IFB0099 infection. When WT Col-0 plants were inoculated with D. dadantii 3937 strain, the most severe disease symptoms developed when plants were grown under an optimal Fe concentration while the opposite trend was detected for both Arabidopsis mutants with impaired biosynthesis of coumarins. During the D. dadantii 3937 infection, both Arabidopsis mutant lines showed a tendency to exhibit more severe disease symptoms in Fe-deficient hydroponics and not in the Fe-sufficient solution as WT Col-0 plants. Although the authors had put much effort into conducting and analyzing data, there are still several concerns about their research methodology and interpretation of results. It seems that normal Fe levels either in root cells or in bacterial cells is associated with more severe symptoms. The authors should provide at least one possible explanation on why higher Fe levels may cause more severe symptoms. One plausible mechanism of symptom enhancement by Fe is its participation in the Fenton reaction where Fe and hydrogen peroxide generates the hydroxyl radical (OH.), one of the most toxic reactive oxygen species. Authors should elaborate this issue in the Discussion.
1/ My specific comment is use of a single reference in roots (EF-1α) and another one in leaf (ACTIN2) for RT-qPCR expression analysis. According to the MIQE (Minimum Information for Publication of Quantitative Real-Time PCR Experiments) guideline, normalization against a single reference gene is not acceptable unless the investigators present clear evidence for the reviewers that confirms its invariant expression under the experimental conditions described. The authors provide no evidence that either reference gene used in this work are stably expressed under the tested conditions (Fe deficiency, pathogen infection).
2/ Lines 275-278 Why these stress marker genes were selected?
3/ I am missing from the manuscript a good summary where they clearly present their results in a few sentences. The most relevant results can be gathered only from many different sites in the manuscript text, but it would be beneficial to find these at a single text site, for example at the end of the abstract or in the conclusions.
Round 2
Reviewer 2 Report
Izabela Perkowska and co-authors, have satisfactorily revised and improved the manuscript. Thank you for their efforts and quick response.